# Research on the Collaborative Evolution of Blockchain Industry Ecosystems in Terms of Value Co-Creation

**Hui Zhang \*, Jin-Biao Yi and Qian Wang**

School of Management, Hangzhou Dianzi University, Hangzhou 310018, China; rencontrer24@163.com (J.-B.Y.); wxixi@hdu.edu.cn (Q.W.)

\* Correspondence: zhanghui816@hdu.edu.cn; Tel.: +86-135-8880-3990

**Abstract:** The formation of blockchain industrial ecology can help improve the open and efficient value synergy network, and this paper seeks to clarify the value relationship among industrial units and the trend of synergistic evolution of blockchain industrial ecosystem. Based on value co-creation theory, the blockchain industrial ecosystem synergy evolution process is analyzed, and the composite system synergy model is used to empirically investigate the evolution synergy of China's blockchain industrial ecosystem from 2015 to 2020. The results show that: although the development level of China's blockchain industry ecosystem continues to improve, it is still at a low level, and the policy-driven effect is obvious. There is a large disparity in the orderliness level of each subsystem of the blockchain industry ecosystem, and the industrial integration and application implementation are in a good situation, while the blockchain enterprises, located in the value pivot subsystem, are in a relatively tough position, and China's blockchain industry ecosystem is overall in a state of reconciliation and has not yet formed a synergistic effect. Handling the synergistic relationship between the government, the market and other value-supporting units, and blockchain enterprises is the top priority for further promoting the synergistic evolution of the blockchain industry ecosystem.

**Keywords:** blockchain; value co-creation; collaborative evolution; industry ecosystem

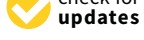



## 1. Introduction

The blockchain industry, an emerging digital technology industry, has received attention from enterprises and governments in various countries due to its characteristic advantages, such as decentralization, openness, peer-to-peer (P2P) networking, and consensus mechanism. The philosophy of synergy, transparency, sharing, and cross-border contained in blockchain provides a new paradigm for building industrial ecology [1], also contributes to the formation and development of industrial clusters, and serves the transformation and upgrading of traditional industries. Since 2016, China has listed blockchain technology as a strategic frontier technology that needs to be laid out ahead of time in the 13th Five-Year National Informatization Plan, and the Chinese government has continuously emphasized the need to strengthen the basic R&D and frontier layout of blockchain in terms of technology, industry, and application. With the guidance of national policies, breakthroughs in basic technologies, and rising demand for application fields, the scale of China's blockchain industry has continued to grow at a high rate, with the number of blockchain enterprises registered surging from 1670 in 2016 to 24,687 in 2020 and the number of blockchain patent applications accounting for 63.52% of the global blockchain patent applications, and the integration of real industries with blockchain technology has been continuously enhanced. The blockchain industrial ecology has taken its initial shape. However, China's blockchain industry is also facing many problems in the development process. Most blockchain enterprises lack support from stable upstream and downstream enterprises, supporting institutions, and other related units in the development and operation process [2]. A benign industrial development environment is needed. Moreover, blockchain technology shows obvious "exclusion reaction" in the process of integration and

development with industry, and the "intrusion" of blockchain has caused a certain impact on the original industrial structure and development operation mode of the region [3] and also caused changes in the industrial policy system, market structure, and industrial environment. Without timely guidance or adjustment, these problems not only influence the stability of the original industry ecosystem value transfer but also lead to the inefficient overall value creation of the blockchain industry. A benignly developing industry should be able to make efficient use of internal and external conditions, improve resource utilization efficiency, perfectly promote an industry synergy and a sound industrial chain [4], achieve value co-creation, and, finally, and realize the healthy development of industrial ecology. Therefore, comprehensively grasping the overall development trend of China's blockchain industry, clarifying the value relationship between industrial units, and figuring out the evolution trend of blockchain industry ecosystem development is undoubtedly beneficial to further expanding the value depth and breadth of China's blockchain industry and strengthen the industry chain synergy.

Value co-creation, a new paradigm of value creation, is a dynamic process in which value units adjust the original and relatively closed operation mode, open the interaction boundary, realize the value interaction between units at all levels [5], and create values together through synergy and resource integration. With the advent of the digital economy era, the emergence of blockchain, big data, artificial intelligence, 5G, and other technologies provides an excellent external environment for the deep integration development and value creation of the industry [6], improving the open, and efficient value exchange network [7]. It enables different units in the system to have a positive impact on value proposition and operational consensus in the process of open interaction and synergy [8], thus promoting the value proposition fit and multisubject symbiosis and adding a solid value power to the high-quality development of the industrial economy. The collaborative evolution of the industry ecosystem is further accelerated by digital empowerment, which makes the system co-creation of value shift from exchange value to social value. Previous research on value co-creation has focused on the two-way interaction between enterprises and consumers [9]. However, due to further development and the complexity of the network economy, the focus of this research has gradually extended to a broader and more diverse perspective concerning the service ecosystem and the interactions among multiple socioeconomic players [10,11]. The value activity vector also extends from a single-value chain to multiple-value networks, becoming more focused on internal value links and synergy of complementary elements. Synergy rules become the key to maintaining the interaction between units in the value network [12]. The value units evolve into more complex and dynamically coupled network interactions [13], which jointly complete the value co-creation process through resource integration, sharing, module decomposition, and innovation synergy [14]. With effective coordination in the industrial ecological environment, a higher level of synergistic development is achieved, which in turn drives the system to realize the dynamic cycle from disorder to order and the evolution of development from a low level to a high level.

The current academic research on the interaction between industry ecosystems and value co-creation has become increasingly refined. The study of value co-creation from the ecosystem level is more compatible with the changes of the current complex and diversified environment [15] and more in line with the needs of theoretical and practical development. However, few scholars have combined the above theories with research regarding the blockchain industry. In terms of the existing literature, the current academic research on blockchain typically focuses on the blockchain technology itself [16], as well as the underlying mechanism of its application to industrial development [6]. While some research focuses on exploring the specific application areas of blockchain technology [17,18], there are limited studies on the overall level of the blockchain industry. Furthermore, there is a lack of macroscopic knowledge about the synergistic development of the blockchain industry. The relevant industry-level studies are generally qualitative and tend to focus on blockchain industry governance [19], operation mechanism [20,21], and

other aspects. These studies are often conducted from a static, meso-, or micro-perspective on the blockchain industry. They also tend to lack quantitative definitive analysis and more in-depth systematic theoretical analysis support, neglecting the complex, self-sustaining, synergistic evolutionary process of the blockchain industry ecosystem, and the basic relationship of value acquisition among units. Based on the value co-creation theory, this paper uses the system synergy model to reveal the degree of synergistic development in China's blockchain industry ecosystem from the analysis of the composition and synergistic evolutionary process of the blockchain industry ecosystem. This paper also aims to grasp the trend of the development and evolution of China's blockchain industry ecosystem to provide some guidance for further expanding the depth and breadth of the value of the blockchain industry and promoting the synergistic development of the industry.

## 2. Analysis of the Composition and Evolutionary Process of Blockchain Industry Ecosystem in Terms of Value Co-Creation

### 2.1. Blockchain Industry Ecosystem Composition in Terms of Value Co-Creation

Early research on industry ecosystems was mainly based on the circular economy perspective, emphasizing the effective circulation of information, materials, and energy within the system, thus achieving an efficient economy and harmonious ecology [22]. Li and Lin [23] defined industry ecosystems from a system viewpoint, arguing that they are formed by the coupling of economic, social, and ecosystem elements that significantly influence industrial development and the synergistic relationship between elements. This enriches the research perspective of industry ecosystems. Han [24] argued that industry ecosystems have the characteristics of open synergy and all stakeholders in the system are potential units of the value co-creation process [25]. The creation and acquisition of values in an ecosystem require the participation of a wider range of units [26]. When building a value co-creation environment, it is crucial to clarify the roles of different units [27]. The degree of trust and synergy between firms [28], as well as influences from environmental factors, such as the market and government [29], can largely determine the value of co-creation behaviors for different units. Chandler and Vargo [30] argued that the system relies on the interaction and synergy of micro-, meso-, and macro-levels to accomplish value creation, which lays the theoretical foundation for constructing the ecosystem value co-creation concept model. Based on the relevant research results and the current situation of the blockchain industry in China, this paper defines the blockchain industry ecosystem in terms of value co-creation. Thus, the blockchain industry ecosystem is a complex system, based on blockchain technology, consisting of three types of value units: (1) the value pivot subsystem; (2) the value support subsystem; and (3) the value integration subsystem, as well as the technical environment, market environment, policy environment, and the user organization of the region where they are located. The value units of each subsystem are interdependent and evolve in a complex system around the development of the blockchain industry. The composition of the blockchain industry ecosystem is shown in Figure 1.

In the blockchain industry ecosystem, the system units interact collaboratively around value creation and value acquisition by embedding in the open and symbiotic ecological environment at corresponding levels and further consolidate and improve value consensus and value activities under the influence of one or more value pivot units [31], which eventually form a benign and healthy value exchange ecology and achieve co-evolution with the system. Among them, blockchain enterprises, blockchain platforms, and user organizations located in the value pivot subsystem are the core, and they are also the key links for value co-creation and value enhancement in the blockchain industry [32]. Out of their value demands, the value pivot units carry out the division of labor and cooperate around the blockchain technology innovation chain and application chain, effectively connecting, realizing interdependence and collaborative creation, accelerating the development and application of blockchain technology, and then realizing the deep integration of blockchain technology with financial services, Internet of Things, supply chain management and other fields within the system, and deriving the value integration subsystem to realize the perfection of industrial ecology and value creation. The value support subsystem is composed of

government agencies, financial institutions, universities and scientific research institutions, regulatory agencies, and third-party intermediary service institutions, etc. The government creates a favorable environment conducive to the research and development of blockchain technology and the promotion and application of products by playing an important role in policy guidance and collaborative supervision [33], reasonably formulating a series of industrial policies such as finance and taxation, talents, and innovation, and strengthening the protection of intellectual property rights and the construction of technical standard systems, etc. The interaction between financial institutions and the value pivot layer can provide financial support and services for enterprises, while the advanced application results provided by blockchain enterprises can realize the decentralized transmission of note values [34], reduce the operational risks brought by the original centralization, greatly enhance the operational efficiency of the whole industrial ecosystem, and realize the gradual upgrading of industrial values. Universities and research institutions provide theoretical methods and talent support for industrial blockchain innovation activities, while science and technology intermediaries provide specialized chain-building and chain-up services to assist each value body to complete the value connection of the industrial chain, which helps the blockchain industrial ecosystem to form a symbiotic and win–win close relationship.

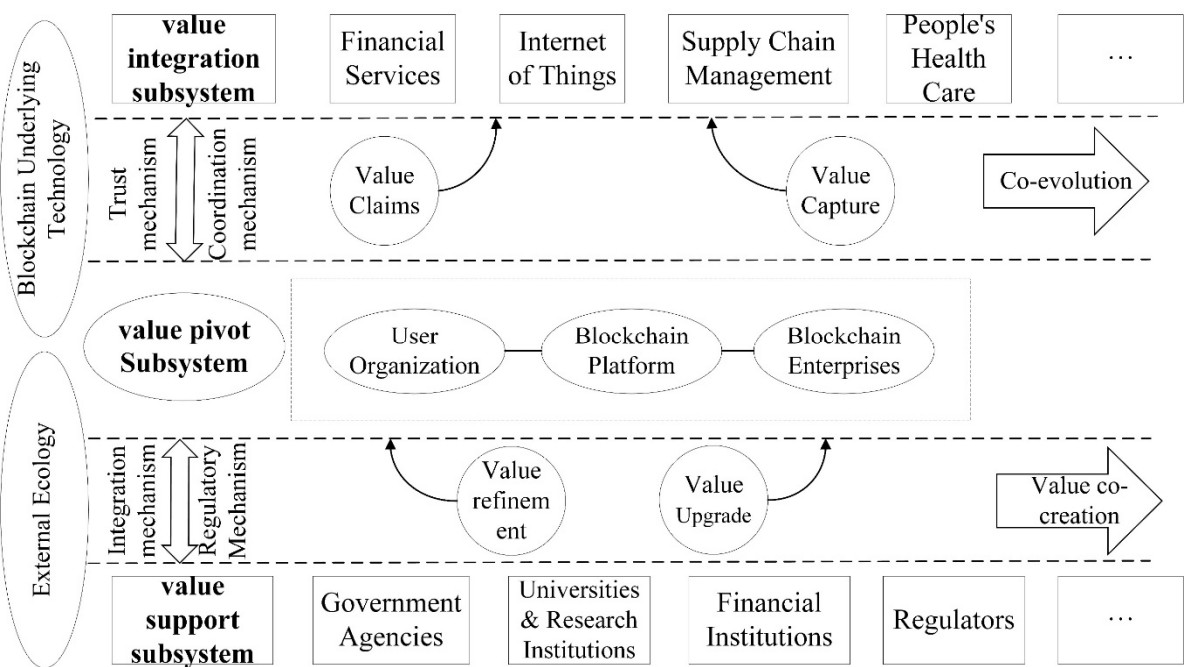

**Figure 1.** Blockchain industry ecosystem composition and synergistic relationship.

## 2.2. Blockchain Industry Ecosystem Composition in Terms of Value Co-Creation

By analyzing the synergistic evolutionary process of the blockchain industry ecosystem, we can sort out the system development process and grasp the value orientation and dynamic relationship of the synergistic development of the system. In previous studies, Wang [35] and Luo [36] defined evolution as the ability to generate value-creating changes within a system that has heredity. Jain and Kogut [10] considered the evolutionary development of the ecosystem as the generation of new valuable functions. According to Jain, its synergistic evolutionary behavior would be influenced by the joint creation of values by system units [37]. The evolutionary development of the blockchain industry ecosystem is the result of mutual influence and a synergistic coupling of system units around value creation and value acquisition in interactions with the internal and external environment, resulting in the realization of value symbiosis [38]. Based on the basic relationship of value formation among system units, its synergistic evolutionary process can be divided into three stages (see Figure 2).

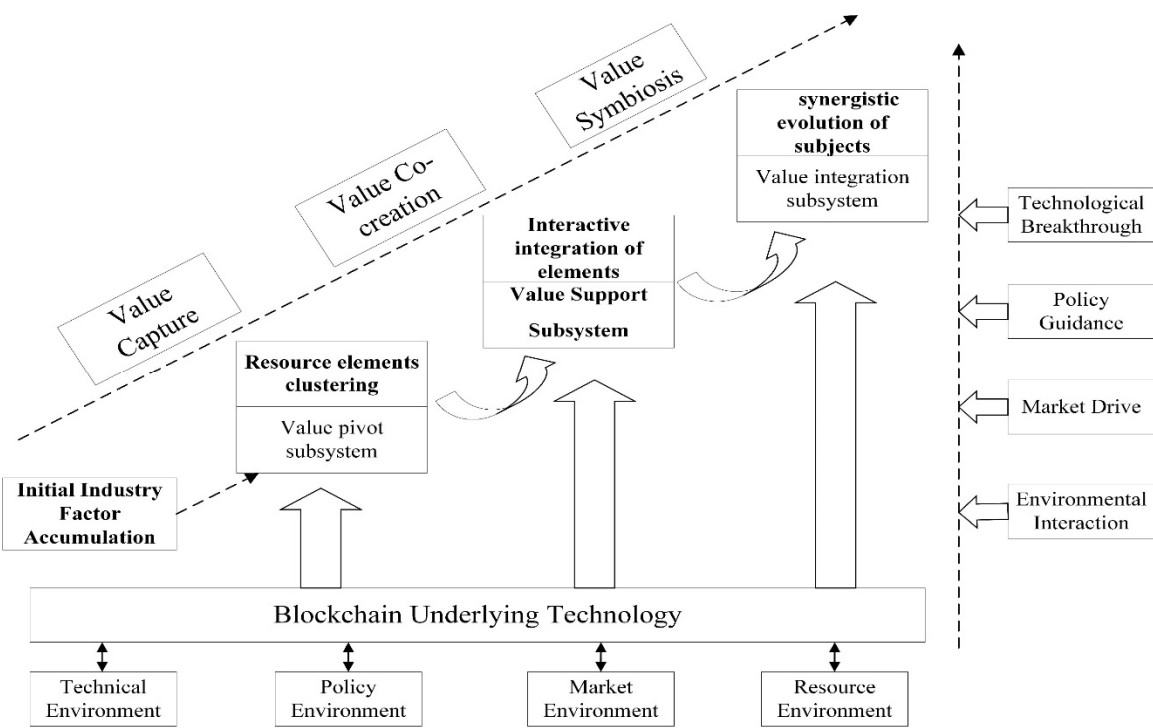

**Figure 2.** Blockchain industry ecosystem synergistic evolution model.

### 2.2.1. Ecopopulation Formation Stage: Aggregation of Resource Elements

The initial stage of the blockchain industry usually suffers from extremely inadequate and unbalanced industrial development factors, such as policy systems, technologies, capital, and talents. First, the single-function blockchain enterprises (individuals) exist independently from the demand for resources and the ability to enhance their value acquisition. Driven by external ecological factors—policy, market, technology, and organizational environment—industrial elements and resources gather rapidly, and the number of blockchain enterprises (species) of the same type accumulate to form blockchain enterprise groups (populations). Through a purposeful exchange of value propositions between different populations, they are linked to each other to form relationships with complementary links, mutual benefits, and synergistic evolution [8]. The modularity and interactivity of blockchain technology itself further enhance the communication and collaboration among system value units, enabling dynamic feedback from different populations to be quickly focused and efficiently responded to [39,40], which makes it possible to create and improve value based on user needs. Blockchain enterprises, on the other hand, by introducing new technologies and products through R&D, building/accessing blockchain open platforms, recruiting partners to deploy alliance chains, and relying on the advantages of the main chain, constitute a value pivot subsystem together with user organizations to develop and build a full range of blockchain application scenarios. These measures are the starting points of value co-creation and sharing, as well as value network hub nodes.

### 2.2.2. Ecocommunity Formation Stage: Interactive Integration of Elements

The accumulation of elements at the early stage of the industry development creates conditions for further integrating system resources and forming blockchain industry communities. Guided by factors such as technological breakthrough, market drive, and policy system, blockchain enterprise populations begin to concentrate and develop in specific regions, forming blockchain enterprise zones (clusters) with stable structures and sound functions and deriving corresponding value support subsystems. These components guarantee the effective circulation of technology, capital, information, and other elements in the system by establishing integration mechanisms, cooperation and innovation mechanisms,

and consensus mechanisms to facilitate the production and upgrade values. At this point, to maximize the satisfaction of their value demands, the enterprises in the community begin to compete and collaborate with similar enterprises, resulting in unbalanced changes in the types of populations, resources, policies, and demands in the blockchain enterprise community. With the deep integration of traditional entities and blockchain technology, product boundaries and organizational boundaries are no longer clear [41,42], and the blurred industrial boundaries promote the deepening application and iterative upgrading of blockchain technology and at the same time promote the digital transformation of traditional industries. Eventually, each population occupies different ecological positions in each link of the value network and form a tacit "mating structure" [43], further strengthening the network effect of the blockchain industrial ecosystem, which in turn meet the heterogeneous needs of different subsystems [44], such that the blockchain industrial community reflects a strong and all-round overall competitiveness.

### 2.2.3. Ecosystem Formation Stage: Synergistic Evolution of Value Units

Due to the collaboration between multiple clusters, the integration and development between the value units are further deepened, and they interact with the internal and external environment. Furthermore, value integration subsystems are derived within the ecosystem, covering many fields, such as financial services, property rights protection, Internet of Things, medical and livelihood, and supply chain management. Through data communication and feedback, the overall efficiency of the value units in collaboration is improved. This further enriches the openness of the blockchain industry system and the scope of value co-creation. The introduction of blockchain technology gives each subject a digital identity [45], which allows the blockchain industry ecosystem to be significantly expandable, and at the same time, it can continuously cultivate new scenarios and new business models for blockchain applications by optimizing the configuration and combination of elements in the system, thus generating new momentum for industry maturity. The synergistic interaction of each subject with technology, information, data, knowledge, and so forth, reaches new heights, which in turn promotes the circulation of values in the blockchain industry ecosystem. This is the ultimate carrier of value co-creation. Under the continuous synergistic evolution of each value subsystem, the system self-organizes to form a more flexible, diverse, and environmentally unrestricted blockchain industry ecosystem. The units in the system are distributed in independent yet interdependent subsystems, which provide great flexibility while ensuring the integrity of the system [46]. The final formation of the industrial ecosystem makes each subject pay more attention to the sum of the system value creation, the dynamics of the process of value transfer and value acquisition, and the nonlinear interactions between value units [47]. When the function of a subsystem is upgraded or changed, the relative balance of dynamic capabilities regulates other value subsystems to complete synergistic matching and maintain the synergistic development among subsystems [35]. This maximizes the value of the blockchain industry ecosystem and, finally, works to realize the overall value symbiosis of the system.

The formation and development of the blockchain industry ecosystem are a dynamic and collaborative evolutionary process. Through the synergistic interaction between the three subsystems of value pivot, value support, and value integration, as well as the uninterrupted information exchange with the internal and external environment, the continuous adjustment and improvement of value demands and industrial ecology are achieved, thus promoting the orderly development of the blockchain industry ecosystem. The orderliness of each subsystem determines whether the evolution of the blockchain industry ecosystem can produce a synergistic effect of 1 + 1 + 1 > 3. Given the synergistic characteristics of the value evolution process of the blockchain industry ecosystem, this paper constructs a blockchain industry ecosystem synergistic development index system. In addition, this paper introduces the system synergy degree model, combined with the relevant data of China's blockchain industry from 2015 to 2020, for empirical analysis to dynamically

test the current situation of the synergistic development of China's blockchain industry ecosystem. It can provide some reference for promoting the integration development of the blockchain industry and the industrial ecology.

## 3. Model Selection and Index System Construction of Blockchain Industry Ecosystem Collaborative Evolution

### 3.1. Composite System Synergy Model

Different methods of synergy measurement have different focuses and different applications. Considering that the blockchain industry ecosystem is a nonlinear system with complex interactions, interpenetrations, and associations among subsystems and elements within subsystems, many factors influence the synergistic development of the three, rendering the synergistic mechanism quite complex. This paper refers to Shen [48] and Wang [49], who chose the composite system synergy model to measure the level of synergistic development of the blockchain industry ecosystem. Let the blockchain industry ecosystem be $S$, $S = \{S_1, S_2, \cdots S_j\}$, $S_j$ is the $j$th subsystem of system $S$, $j \in [1,3]$, among which the value integration subsystem is $S_1$, the value pivot subsystem is $S_2$, and the value support subsystem is $S_3$. The compound mechanism of the blockchain industry ecosystem $S$ is formed by the interaction and mutual influence among subsystems $S_j$.

### 3.1.1. System Orderliness Model

Let the order parameter of the subsystem $S_j$ in the development process be $e_j = (e_{j1}, e_{j2}, \cdots, e_{jn})$, $n \geq 1$, $\alpha_{ji} \leq e_{ji} \leq \beta_{ji}$, $i \in [1, n]$. Assuming that $e_{j1}, e_{j2}, \cdots, e_{jk}$ are positive indicators, then the larger the order parameter $e_j$, the higher the orderliness of the system, and vice versa. Assuming that $e_{jk+1}, \cdots, e_{jn}$ are negative indicators, then the larger this value, the lower the orderliness of the system, and vice versa. Thus, the order parameter component $e_{ji}$ of subsystem $S_j$ is ordered as follows:

$$u_j(e_{ji}) = \begin{cases} \frac{e_{ji} - \alpha_{ji}}{\beta_{ji} - \alpha_{ji}} i \in [1, k] \\ \frac{\beta_{ji} - e_{ji}}{\beta_{ji} - \alpha_{ji}} i \in [k+1, n] \end{cases} \tag{1}$$

where $u_j(e_{ji}) \in [0, 1]$; $\alpha_{ji}$ and $\beta_{ji}$ are the upper and lower limits of the order parameter $e_{ji}$ of the $j$th subsystem in the steady-state, respectively. A larger $u_j(e_{ji})$ value indicates that the "contribution" of $e_{ji}$ to the orderliness of the system is greater.

The orderliness of the order parameter $e_j$ to the subsystem $S_j$ can be calculated by the geometric mean or linear weighted summation method as follows:

$$u_j(S_j) = \sqrt[n]{\prod_{i=1}^{n} u_j(e_{ji})} \tag{2}$$

or

$$u_j(S_j) = \sum_{i=1}^{n} \omega_i u_j(e_{ji}), \omega \geq 0, \sum_{i=1}^{n} \omega_i = 1 \tag{3}$$

From Equations (2) and (3), we can see that for $u_j(S_j) \in [0, 1]$, the larger this value, the higher the orderliness of the subsystem, and vice versa.

### 3.1.2. System Synergy Model

If at a given moment, $t_0$, the orderliness of each subsystem ordinal parameter is $u_j^0(s_j)$, $j = 1$, 2, 3; then when the composite system runs to moment $t_1$, the orderliness of each subsystem ordinal parameter is $u_j^1(s_j)$; and, at this time, the synergy between the two subsystems within the composite system is modeled as

$$c_m(s_h, s_k) = \frac{1}{e-1} \left[ e - \exp(1 - \prod_{j=h,k} u_j^1(s_j) - u_j^0(s_j)) \right] \tag{4}$$

where $u_j^1(s_j) - u_j^0(s_j) \neq 0, c \in [0,1]$, and $c_m(s_h, s_k)$ is positively correlated with the overall composite system synergy. The composite system synergy degree consists of the collection of subsystem synergy degrees, which is modeled as

$$U(S) = \varepsilon \sqrt[3]{\prod_{j=1}^{3} c_m(s_h, s_k)} \qquad \begin{cases} \varepsilon = 1 \ u_j^1(s_j) \geq u_j^0(s_j) \\ \varepsilon = 0 \ u_j^1(s_j) \leq u_j^0(s_j) \end{cases} \tag{5}$$

The parameter $\varepsilon$ can judge the direction of the influence of subsystems on the composite system. When $\varepsilon = 1$, the synergy degree $U(S)$ is positive, indicating that the blockchain industry ecosystem is in a coordinated and orderly development state. The larger the value of $U(S)$, the higher the degree of synergy and orderliness of the system, and vice versa. The degree of synergy for the composite system depends on the common effect of all subsystems—i.e., the low orderliness of a subsystem affects the synergy degree of the system. There is no unified standard for the classification of the grade of the synergy degree of the composite system in academia. In this paper, we set the grade and classification criteria for the overall degree of the synergy of the blockchain industry ecosystem and the degree of synergy among subsystems (see Table 1).

**Table 1.** Synergy level and classification criteria.

| Synergy | 0–0.4 | 0.4–0.5 | 0.5–0.6 | 0.6–0.7 | 0.7–0.8 | 0.8–0.9 | 0.9–1.0 |
|---------|-------|---------|---------|---------|---------|---------|---------|
| Grade Criteria | Disorder | Harmonization | Barely Synergistic | Primary Synergy | Intermediate Synergy | Good Synergy | Superior Synergy |

### 3.2. Construction of the Index System

Considering that China's blockchain industry is at the early stage of development, the relevant data are not yet perfect, and the statistical indexes are not completely consistent. According to the principles of scientific and systematic index selection and data availability, this paper constructs an index system for the blockchain industry ecosystem synergistic development, which includes three subsystems—value pivot, value integration, and value support—with a total of 22 subdivision indexes (see Table 2). The value integration subsystem and value support subsystem adopt the overall level data of China's blockchain industry, while the development of the value pivot subsystem is mainly reflected in blockchain enterprises. Therefore, this paper adopts 240 listed companies in the China A-share blockchain sector as the initial sample after excluding: (1) enterprises with less than six years of listing time; (2) enterprises with no input or output in blockchain; and (3) enterprises in the ST category that are specially treated. The remaining 178 blockchain-listed enterprises are used to reflect the value pivot subsystem.

### 3.3. Determination of Index Weights

The entropy value method is used to determine the coefficients by using the degree of difference between the values of evaluation indicators, which can avoid the bias brought by subjective factors in the process of determining the weight coefficients. It can also reflect the importance of each indicator in the comprehensive index system more objectively. Let $X_{ij}$ be the value of the $j$. th indicator in the $i$-th year, where $i = 1, 2, \cdots, n; j = 1, 2, \cdots, m$. indicates the number of years and m indicates the number of indicators. The specific process is as follows:

1. Dimensionless treatment of indicators.

$$\text{Positive indicator}: \ X'_{ij} = \frac{X_{ij} - X_j^{min}}{X_j^{max} - X_j^{min}} \tag{6}$$

$$\text{Negative indicator}: \ X'_{ij} = \frac{X_j^{max} - X_{ij}}{X_j^{max} - X_j^{min}} \tag{7}$$

2. The authors chose to pan the whole data after dimensionless processing by 0.01 to normalize the index system, to eliminate the 0 values that appear in the process, and, at the same time, to minimize the influence of panning on the original data.

$$P_{ij} = \left( X'_{ij} + 0.01 \right) / \sum_{j=1}^{m} (X'_{ij} + 0.01) \tag{8}$$

**Table 2.** The index system for the blockchain industry ecosystem synergy development.

| System | Subsystems | Sequence Parameters | Sequence Variables | Weights | Data Sources |
|---|---|---|---|---|---|
| Blockchain Industry Ecosystem Synergy Development Index | Value Integration Subsystem | Industry Value Scale | X1: Market Scale | 31.54% | 2020–2021 China Blockchain Industry Development White Paper |
| | | Degree of Industrial Agglomeration | X2: Number of Blockchain Industry Zones | 14.85% | |
| | | Industry Attractiveness | X3: Number of Blockchain Companies | 19.01% | Tian-eye Search Business Inquiry Platform |
| | | Degree of Industrial Integration | X4: Blockchain Application Area Patents | 16.11% | Innojoy Patent Search Platform |
| | | | X5: Number of Blockchain Application Cases | 18.49% | Tsinghua University's 2020–2021 China Blockchain Industry Ecological Map Report |
| | Value Pivot Subsystem | Innovation Capability | Y1: Number of R&D staff | 8.11% | CSMAR Database, Enterprise Annual Reports |
| | | | Y2: R&D expenditure | 11.86% | |
| | | | Y3: Enterprise Research Projects | 14.78% | Pan Research Global Research Project Database |
| | | | Y4: Number of Enterprise Patents | 15.51% | Innojoy Patent Search Platform |
| | | | Y5: Number of Enterprise Academic Papers | 12.71% | Chinese National Knowledge Infrastructure (CNKI) |
| | | Profitability | Y6: Return on Assets | 8.71% | CSMAR Database |
| | | Growth Capability | Y7: Total Assets Growth Rate | 11.68% | |
| | | Solvency | Y8: Property Index | 6.64% | |
| | | Operating Capability | Y9: Asset Turnover Ratio | 10.00% | |
| | Value Support Subsystem | Government Support | Z1: Number of Policies | 14.81% | China Blockchain Industry Development Census Report (2020) |
| | | | Z2: Government Subsidies | 7.65% | Enterprise Annual Reports |
| | | Capital Support | Z3: Number of Financing | 13.28% | China Blockchain Investment and Financing Report (2020) |
| | | | Z4: Amount of Financing | 21.35% | |
| | | Talent Support | Z5: Number of Schools Offering Blockchain Majors | 11.14% | Web Search |
| | | Innovation Support | Z6: Number of Noncompany Research Projects | 10.46% | Pan Research Global Research Project Database |
| | | | Z7: Number of Nonenterprise Blockchain Patents | 11.36% | InnojoyPatent Search Platform |
| | | | Z8: Number of Nonbusiness Research Papers | 9.95% | CNKI |

Note: In order to exclude the interference of "pseudo" blockchain enterprises as far as possible, X3 in the index only includes blockchain enterprises with registered capital of more than 10 million; X4 only includes the number of patents in the fields of finance, Internet, supply chain, healthcare, Internet of Things, and games; X5 refers to the number of blockchain projects actually put into application; enterprise-level indexes X6–X9 are the average values, and the rest are aggregated values; Y3 and Z6 are obtained by searching whether the project undertaking organization includes the word "company"; Y4 and Z7 are obtained by searching whether the patent applicant includes the word "company"; X4 and Z8 are obtained by searching whether the author's X4 and Z8 are retrieved based on whether the authors include the word "company"; Z1 only includes national blockchain policies; and Z5 refers to the number of colleges and universities in mainland China that offer blockchain-related professional courses.

3. Calculate the entropy value of the *j*th indicator.

$$E_j = -\frac{1}{\ln(m)} \sum_{i=1}^{n} (P_{ij} \times lnP_{ij}) \tag{9}$$

4. Calculate the weight of the *j*th indicator.

$$\omega_j = (1 - E_j) / \left( m - \sum_{j=1}^{m} E_j \right) \tag{10}$$

## 4. Measurement and Analysis of the Blockchain Industry Ecosystem Synergy

### 4.1. Subsystem Development Level Measurement

According to the constructed index system for the blockchain industry ecosystem collaborative development, the entropy value method is used to calculate the weight of each index using Matrix Laboratory (MATLAB) 2016b software, and the results are listed in Table 2. The indexes are further linearly weighted to obtain the comprehensive development level score of each subsystem of the blockchain industry ecosystem from 2015 to 2020 (see Table 3 and Figure 3).

**Table 3.** The comprehensive development level of blockchain industry ecosystem subsystems.

| Year | Value Integration Subsystem | Value Pivot Subsystem | Value Support Subsystem |
|------|------|------|------|
| 2015 | 0.0011 | 0.0358 | 0.0022 |
| 2016 | 0.0049 | 0.0402 | 0.0144 |
| 2017 | 0.0188 | 0.0389 | 0.0393 |
| 2018 | 0.0523 | 0.0576 | 0.1477 |
| 2019 | 0.0623 | 0.0690 | 0.0906 |
| 2020 | 0.1116 | 0.0818 | 0.1317 |

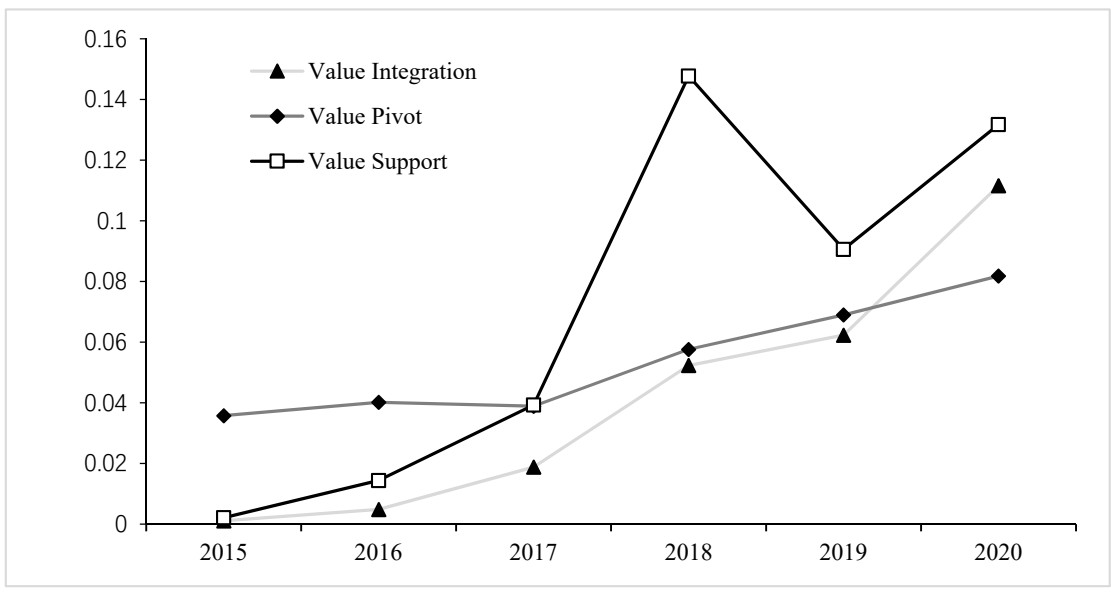

**Figure 3.** Blockchain industry ecosystem subsystem development level in 2015–2020.

Figure 3 and Table 3 show that although the development level of China's blockchain industry ecosystem shows an obvious upward trend, it is still at a low level overall. From the weighting of each sequential covariate in Table 2, the blockchain industry value scale, industry innovation capability, and capital support contribute the most to the evolutionary development of the blockchain industry. In addition, it is obvious from Figure 3 that China's blockchain industry slowly increased in 2015–2016, and a turning point was

reached in 2017. All three major subsystems developed substantially in 2017, among which the value support subsystem has had the greatest growth rate. This is closely related to a series of central policies, such as the "White Paper on the Development of Blockchain Technology and Applications in China" (2016), issued by the Ministry of Industry and Information Technology in October 2016, and the "Notice of the State Council on the Issuance of the 13th Five-Year National Informatization Plan" issued by the State Council in December 2016, in which blockchain was included as a strategic frontier technology for the first time. The fall in the value support subsystem in 2018 may be attributed to the fact that the central government introduced a series of measures to strengthen the regulation of the blockchain industry in 2018. Additionally, the market returned to rationality, while also giving rise to several high-quality projects. In 2020, the "crises" and "opportunities" brought on by the pandemic further highlighted the importance of blockchain and other new-generation information technology, which accelerated the deep integration and application of blockchain technology within various industries. The level of all three subsystems of the blockchain industry has greatly improved in 2020, and the overall development prospect in the postpandemic era is promising.

### 4.2. Subsystem Orderliness Measurement

The dimensionless processed data are substituted into Equation (1), where $\alpha_{ji}$ and $\beta_{ji}$ are taken as the upper and lower limit values of the ordinal variables in 2015–2020 with the maximum value up by 10% and the minimum value down by 10%, respectively. The orderliness of each ordinal variable is calculated, and then the orderliness of the ordinal variable is substituted into Equation (2) to obtain the orderliness of each subsystem of the blockchain industry ecosystem (see Table 4 and Figure 4).

**Table 4.** Blockchain industry ecosystem subsystem degree of orderliness.

| Year | Value Integration Subsystem $U_1(S_X)$ | Value Pivot Subsystem $U_2(S_Y)$ | Value Support Subsystem $U_3(S_Z)$ |
|------|-----------------------------------------|-----------------------------------|-------------------------------------|
| 2015 | 0.0017 | 0.0396 | 0.0030 |
| 2016 | 0.0343 | 0.2379 | 0.0544 |
| 2017 | 0.1644 | 0.3357 | 0.1681 |
| 2018 | 0.4873 | 0.4590 | 0.4640 |
| 2019 | 0.5769 | 0.4421 | 0.4221 |
| 2020 | 0.8937 | 0.2730 | 0.5113 |

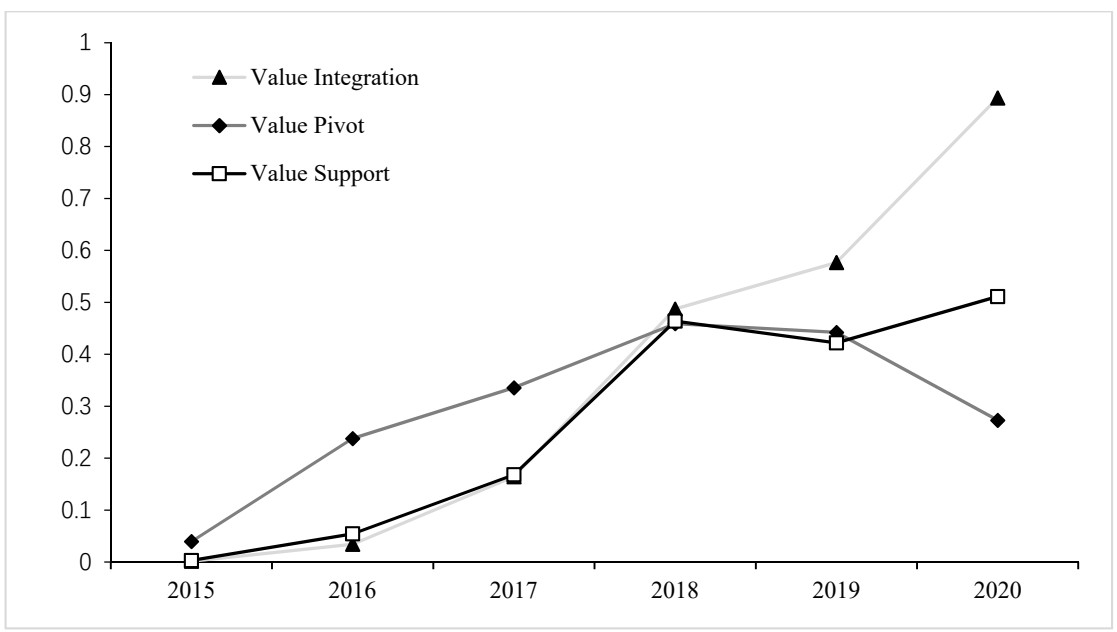

**Figure 4.** Blockchain industry ecosystem subsystem orderliness in 2015–2020.



Figure 4 and Table 4 show that the orderliness level of each subsystem of the blockchain industry ecosystem has maintained approximately the same growth trend from 2015 to 2018. However, differences in the orderliness level of each subsystem emerged in 2018, where the value integration subsystem still maintained a growth trend, and the value pivot subsystem and value support subsystem showed a decrease in orderliness. The main reason for this is the initial massive investment of the blockchain industry, as an emerging industry, in R&D resources and elements, which gave rise to the first wave of the climax of blockchain technology application landing in 2017. However, the return cycle of its application development is relatively long, and the blockchain business of each enterprise has not yet reached the point of self-sufficiency. Other businesses are still needed to cover the cost of blockchain business, thus causing the blockchain industry to bloom in the application field. However, the blockchain enterprises located in the value pivot subsystem are generally in a difficult situation, which in turn reduces the orderliness of the subsystem. The COVID-19 pandemic further exacerbated the dilemma of blockchain companies, and the orderliness level of the value pivot subsystem decreased significantly. By contrast, the value support subsystem, especially the value convergence subsystem, was again accelerated in the pandemic environment, leading to a large gap in the orderliness level of each subsystem.

### 4.3. Overall Synergy Measurement of the Blockchain Industry Ecosystem

The synergy degrees $c_m(s_x, s_y)$, $c_m(s_x, s_z)$, and $c_m(s_y, s_z)$ between the three major subsystems of the blockchain industry ecosystem for 2015–2020 are derived by substituting the ordered degrees of value integration, value pivot, and value support subsystems, calculated in Table 4, into Equation (4). The calculated results are brought into Equation (5) and finally obtain the overall synergy $U(S)$ of the blockchain industry ecosystem in 2015–2020 (see Table 5 and Figure 5).

**Table 5.** Blockchain Dimensionless treatment of indicatindustry ecosystem degree of synergy.

| Year | Value Integration-Value Pivot $c_m(s_x,s_y)$ | Value Integration-Value Support $c_m(s_x,s_z)$ | Value Pivot-Value Support $c_m(s_y,s_z)$ | Overall Synergy $U(S)$ |
|---|---|---|---|---|
| 2015 | 0.0082 | 0.0022 | 0.0109 | 0.0059 |
| 2016 | 0.0804 | 0.0409 | 0.1009 | 0.0692 |
| 2017 | 0.2195 | 0.1639 | 0.2210 | 0.1996 |
| 2018 | 0.4513 | 0.4731 | 0.4396 | 0.4545 |
| 2019 | 0.4812 | 0.4909 | 0.4107 | 0.4595 |
| 2020 | 0.4563 | 0.6733 | 0.3444 | 0.4730 |

Figure 5 and Table 5, along with the synergy level classification criteria, demonstrate that although the synergy of China's blockchain industry ecosystem continued to improve from 2015 to 2017, it was still in a state of dissonance. From 2018, it started to change to a state of reconciliation, but it has remained relatively stable, without achieving further synergy in the system. The reason for this is that there is a "barrel effect" in the coordinated development of the blockchain industry ecosystem; i.e., the "shortboard" of the orderliness of the value pivot subsystem restricts the level of synergistic development for the whole system to a certain extent.

From the dynamic evolutionary process of the synergy trend of China's blockchain industry ecosystem, the overall orderliness of the industry's subsystems maintained a stable growth trend from 2015 to 2017. Therefore, the overall system synergy was continuously improved. In 2018–2020, although the orderliness of the value support subsystem, especially the value integration subsystem, achieved significant growth, the value pivot subsystem orderliness decreased and remained at a low level, resulting in no further improvement in the overall system synergy. Secondly, in terms of the static status quo of the synergistic development of China's blockchain industry ecosystem, although the orderliness of the value support subsystem, especially the value integration subsystem, reached a

high level in 2020, the orderliness level of the value pivot subsystem was at a relatively low level, thus manufacturing the synergistic development of the whole industrial system. Therefore, regardless of the dynamic evolution process of the synergistic evolutionary process of China's blockchain industry ecosystem or the static development status quo, the orderly development of the value pivot subsystem has become a key factor restricting the synergistic development of the whole blockchain industry ecosystem.

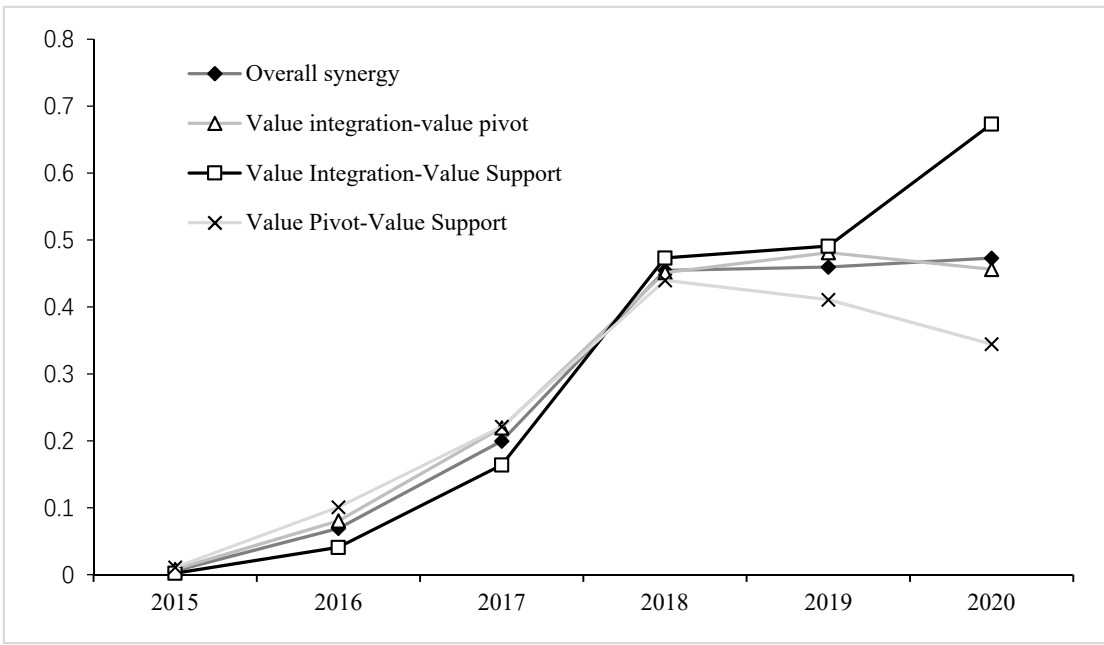

**Figure 5.** Blockchain industry ecosystem synergy trend in 2015–2020.

Further, according to the trend of synergistic evolution between various subsystems of the blockchain industry ecosystem, it can be seen that the synergy between the value integration subsystem and the value support subsystem keeps growing continuously, and the synergy is close to 0.7 in 2020. Additionally, the value pivot subsystem and the value integration subsystem have maintained a relatively stable state since 2018. However, the synergy between the value support subsystem and the value pivot subsystem has continuously decreased from 2018 onward; thus, the overall synergy of the system did not improve further. This implies that the government, market, and other macro units located in the value support subsystem have overly tilted the focus of blockchain industry development to the landing application and value integration of blockchain technology, without coordinating the relationship with blockchain enterprises located in the value pivot subsystem, to further promote the synergistic evolutionary process of China's blockchain industry ecosystem and the orderly development of the value pivot subsystem—especially the value. To further promote the synergistic evolutionary process of China's blockchain industry ecosystem, it is of utmost importance to promote the orderly development of the value pivot subsystem, especially the synergy between the value pivot subsystem and value support subsystem.

## 5. Conclusions and Suggestions

In this paper, by sorting out the studies related to value co-creation and blockchain industry development, as well as the actual situation of China's blockchain industry, we analyzed the composition of the blockchain industry ecosystem and its synergistic evolutionary process from the perspective of value co-creation, and on this basis, we made an empirical analysis of the synergistic degree of China's blockchain industry ecosystem from 2015 to 2020 using the system synergy degree model. The main conclusions are as

follows: (1) the development level of China's blockchain industry ecosystem continues to improve, but it is still at a low level, and the policy-driven effect is very obvious; (2) the disparity in the orderliness level of each subsystem of the blockchain industry ecosystem is large, and the industrial integration and application implementation are in a good situation, but the blockchain enterprises located in the pivotal position of the industry are in a relatively difficult situation; and (3) the overall blockchain industry ecosystem in China is in a reconciliation state and has not formed synergy effects yet. Handling the synergistic relationship between the government, market, and other value-supporting bodies and blockchain enterprises is the most important point to further promote the synergistic evolutionary process of the blockchain industry ecosystem. In this regard, this paper puts forward the following policy suggestions.

Strengthen the top-level design and improve the value ecology. It is necessary to effectively strengthen the top-level design of blockchain technology applications. Relevant departments should play a coordinating and collaborative role to clarify the specific areas of integration between blockchain and the real economy, strengthen the coordination and docking of strategy, technology, standard, market, talents, and other aspects, integrate and optimize resource elements, and give corresponding top-level support. In addition, the government should take the lead in deploying blockchain infrastructure with platforms, standardization, and components and laying out the certification mechanism and standard system for blockchain applications. The government should also assist in guiding industrial entities to complete the chain construction, chain uploading, and chain reform and give full play to the policy-driven effect at the early stage of the industry to promote the scientific, orderly, and coordinated development of the "blockchain+" model and guide the blockchain industry to ecological perfection.

Pay attention to the role of enterprises and convert the value kinetic energy. Blockchain enterprises play a pivotal role in many aspects, such as an industrial–technological breakthrough, application integration, and value creation, while also standing at the forefront of the industry in the face of market fluctuations. Therefore, while coordinating the overall situation of the industry, it is important to start building a new value collaboration and delivery architecture with enterprise organizations as the anchor point and crossdomain interaction and collaboration as the grid. The government cannot be detached from the status quo of enterprises and should shift its policy focus down to the enterprise level to highlight the cultivation of market players, guide market elements to gather in enterprises, and promote the scale and cluster development of blockchain enterprises. Considering the long payback period of blockchain R&D and application and the difficulty and high cost of trial and error for enterprises, the government can appropriately provide financial subsidies for pilot projects of blockchain applications. This reduces the difficulty of starting enterprise blockchain projects and trial and error costs and accelerates the marketization and scale of the industry and guide the development of the blockchain industry from policy single-core driven to policy + market dual-core driven.

Improve market regulation and promotion of value synergy. A sound regulatory mechanism guarantees the integration and development of blockchain and various industries. The government should strengthen the innovation of the regulatory model, adopt the concept of cooperative regulation, give full play to the regulatory functions of other value units, and jointly improve the regulatory system of the blockchain industry. In addition, it should strengthen the whole life-cycle management of blockchain platform, determine the potential risks of the platform, adjust and improve the regulatory measures in the field of integration of blockchain and the real economy, suggest a trust-based regulatory system, implement "rule the chain with the chain", and make up for the regulatory shortcomings with technical advantages. A sound regulatory system is used to create a rational and healthy industrial development environment, reduce capital bubbles and market fluctuations in the blockchain industry, and achieve benign synergy and value co-creation of multidimensional value units, such as government, market, and enterprises.

There are still some limitations in this paper: firstly, the conceptual model of blockchain industrial ecosystem composition and the analysis framework of synergistic evolution need to be further deepened. This paper constructs the structure of the blockchain industry ecosystem from the perspective of value co-creation and analyzes and verifies its synergistic evolution process, but the conceptual model and analytical framework constructed in this paper only provide a preliminary theoretical perspective and foundation for the research on the blockchain industry ecosystem. Its deeper theoretical and practical significance still needs further research to explore and realize gradually. Secondly, due to the lack of statistical data and availability of the existing blockchain industry, the indicators selected in this paper may not be comprehensive enough and lack of specificity. Since the blockchain industry is still in the early stage of development, the existing statistical caliber is not perfect, and the ambiguity of the blockchain industry boundary itself also brings difficulties to the collection of relevant data, and more comprehensive and accurate statistics are needed to enrich the empirical evidence. Future research can enrich and improve the research related to the blockchain industry ecosystem in terms of case study, formation mechanism, influencing factors, simulation analysis, etc.

**Author Contributions:** Conceptualization, H.Z. and J.-B.Y.; data curation, Q.W.; formal analysis, H.Z. and J.-B.Y.; funding acquisition, H.Z.; investigation, H.Z. and J.-B.Y.; methodology, J.-B.Y. and Q.W.; project administration, H.Z. and Q.W.; resources, J.-B.Y.; software, J.-B.Y.; supervision, J.-B.Y.; validation, H.Z. and Q.W.; visualization, Q.W.; writing—original draft, J.-B.Y.; writing—review and editing, H.Z. and Q.W. All authors have read and agreed to the published version of the manuscript.

**Funding:** This research was funded by the National Social Science Fund of China, grant number 20BGL293.

**Institutional Review Board Statement:** Not applicable.

**Informed Consent Statement:** Not applicable.

**Data Availability Statement:** Not applicable.

**Conflicts of Interest:** The authors declare no conflict of interest.

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
