# Peer review of "Research on the Collaborative Evolution of Blockchain Industry Ecosystems in Terms of Value Co-Creation"

_sustainability, doi:10.3390/su132111567_

Round 1
Reviewer 1 Report
I recommend the authors broaden the discussion in the introduction highlighting the contextualization and motivations of this study. Why do the authors want to carry out this study in the context of China? What are the special characteristics and economic implications of this city that motivated the authors to undertake this exercise?
The most relevant literature on the topic at stake is contained in the paper and the most relevant theories stemming from that literature are clearly expressed. Yet, I recommend some other relevant papers to include:
- Wolfond, G. (2017). A blockchain ecosystem for digital identity: improving service delivery in Canada’s public and private sectors. Technology Innovation Management Review, 7(10).
- Cicchiello, A. F. (2019). Building an entrepreneurial ecosystem based on crowdfunding in Europe: the role of public policy. Journal of Entrepreneurship and Public Policy.
Authors should pay more attention to the clarity of expression and readability. Some of the sentence construction needs to be clearer.
Reviewer 2 Report
This is excellent research that assesses the contribution provided by the introduction and evolution of the Blockchain in the industries in China.
The theoretical framework used is value co-creation and is further supported by a synergy model implemented by the authors for this purpose.
I would like to suggest only the following
- Include the research question in the abstract
- better support and describe figure 1 and Figure 2, not adequately explained in the text
- The choice of the Sequence variables is not disclosed/explained (weights attributed through entropy method).
- Limitations and further research development of the research not identified (they are expected to be disclosed in the "conclusion and suggestion" section)
